# Therapy of Extensive Chronic Skin Defects after a Traumatic Injury Due to Microbial Contamination Using a Surface Implant Made of a Biocompatible Polycaprolactone—A Pilot Case Study

**DOI:** 10.3390/polym14235293

**Published:** 2022-12-03

**Authors:** Alena Findrik Balogová, Martin Kožár, Radka Staroňová, Marek Schnitzer, Gabriela Dancáková, Jozef Živčák, Radovan Hudák

**Affiliations:** 1Department of Biomedical Engineering and Measurement, Faculty of Mechanical Engineering, Technical University of Košice, 042 00 Košice, Slovakia; 2Small Animal Clinic, University Veterinary Hospital, University of Veterinary Medicine and Pharmacy in Košice, Komenského 73, 041 81 Košice, Slovakia

**Keywords:** polycaprolactone, additive manufacturing, soft tissue treatment, skin defect, scaffold

## Abstract

This case study describes the use of additive manufacturing technology combining a biodegradable polymer material, polycaprolactone (PCL), and innovative procedures for creating superficial wound dressing, a scaffold in the therapy of extensive contaminated skin defects caused by a traumatic injury. Chronic and contaminated wounds represent a clinical problem and require intensive wound care. The application of a temporary scaffold-facilitated bridging of the wound edges resulted in faster tissue regeneration and a shorter defect closure time, compared to other conservative and surgical methods used in therapy of chronic wounds. Although this procedure has proven to be an optimal alternative to autologous transplants, further studies with a larger number of patients would be beneficial.

## 1. Introduction

Soft tissue healing is a process of repairing various structures, with the aim of achieving tissue regeneration. The most frequent form of tissue reparation is closing the damaged area with scar tissue; this, however, is associated with a partial loss of function. Another form of healing is tissue regeneration in which tissues are newly formed and achieve the same condition as before the injury, with only minimal, or no, loss of function. In an ideal case, tissue regeneration is the primary organism’s response to damage. Unfortunately, in most cases, the healing of tissues of living organisms after an injury or damage is associated with the formation of scars. It is, therefore, recommended that the therapeutic management includes influencing the individual stages of the healing process, in order to recover the function of the affected site to the greatest possible extent.

Wound healing consists of several consecutive stages. Each stage is characterised by specific biological processes. The early stage of healing includes inflammation and haemostasis. The granulation stage consists of proliferation of mesenchymal cells, migration, epithelisation, and angiogenesis. The basic processes in later stages are the synthesis of collagen and other matrix proteins, as well as wound contraction. Scar remodelling is the basic characteristic feature of the final stage of the healing process. The individual stages overlap and have no clear borders [1,2]. 

Traumatic injuries of pet animals, most frequently dogs and cats, that result in large open wounds may be accompanied by complications related to tissue dislocations, soft tissue, and skin infections, as well as a subsequent transition to the chronic stage. One of the causes of complications in chronic wounds is the development of a secondary bacterial infection of soft tissues, where aseptic and antiseptic conditions are neglected within therapeutic procedures [3]. The most severe bacterial infections that occur in such cases include the infections caused by *Staphylococcus pseudointermedius*, *Staphylococcus aureus*, or *Staphylococcus haemolyticus* strains [4,5,6,7]. If those strains are resistant, therapy is mostly complicated [8]. Chronic defects are not rare. They mostly affect the surrounding soft tissues, including the skin. Healing of such defects, therefore, discontinues because the strains of methicillin-resistant Staphylococcus aureus (MRSA), such as *Staphylococcus aureus* or *Staphylococcus haemolyticus*, produce toxins that cause deterioration and retardation of the healing process, persistence of the inflammation stage, and necrotising of the adjacent tissues [9,10]. All of these factors result in the development of extensive necrotic wounds that are difficult to treat with conservative methods.

Therefore, the most serious complication in the process of healing infected wounds is the transition to chronicity. A chronic wound is defined as a wound that fails to heal over three weeks with standard wound care [11]. Moreover, chronic wounds require intensive standard wound care; however, this is sometimes insufficient to promote healing [12].

One of the ways to accelerate the regeneration of the surrounding soft tissues is the application of innovative procedures, such as the fusion of biodegradable, biocompatible polymer materials, and additive manufacturing technology [13,14,15]. The purpose of this therapy was to create an artificial bridge over the wound—a scaffold. The expected result of using a scaffold made of a polymer biocompatible material as temporary wound dressing was bridging the wound edges. Another purpose of applying this scaffold directly onto the surface of the skin defect was to drain off wound exudates, provide access of the air to the wound, and create new tissues on the skin, with the consequent aim of reducing the risk of infections of subdermal structures.

Polycaprolactone could be one of the suitable materials that can be used for this purpose. Polycaprolactone is a polymeric biocompatible material that, in its medical degree of purity, is used for various purposes in human medicine, too [16]. Due to its optimal mechanical properties, polycaprolactone has great biopotential in the biomedical applications of tissue engineering [17]. Several studies also refer to the choice of polycaprolactone as a suitable material for use in the present case study [17,18] and mechanical properties [19,20].

## 2. Materials and Methods

### 2.1. Materials

Polycaprolactone (PCL) Purasorb PC 12, supplied in form of granules by Corbion Purac (Amsterdam, The Netherlands). The seller declares: “Biodegradable and biobased, our PURASORB^®^ polymers promote natural wound management and healing via safe and effective surgical and other fiber-based products. They are extremely flexible and can be used in synthetic resorbable mono- and multi-filaments” [21]. The Purasorb PC 12 is primarily used for medical devices and applications [22]. Using the Devo equipment: Filament maker (3Devo, Utrecht, The Netherlands), the granules were processed into a filament with a diameter of 1.75 mm. The scaffold, i.e., a temporary implant, was designed and created in the Magics environment (Materialise, Plymouth, MI, USA). The 3D printing of the final scaffold was carried out using the DeltiQ 2 Plus printer (TriLAB, Hradec Kralove, Czech Republic).

#### 2.1.1. Sterilisation of Temporary Implants

Primary sterilization with isopropyl at the 3D printing site. The sterilization process was repeated in the hospital shortly before the application of the temporary implants. Solution of chlorhexidine (Skinmed Chlorhexidin Spray, Cymedica, Czech Republic) in a 4% concentration (Aqua pro injectione, Bieffe Medital S.p.a., Grosotto, Italy) was used for immersing the implant for 10 min. Subsequently, the implant was rinsed with sterile physiological solution (B. Braun, Melsungen, Germany).

#### 2.1.2. Anaesthesia

Butomidor containing butorphanol active substance (Richter Pharma, Wels, Austria) at a dose of 0.25 mg/kg of live mass, combined with Cepetor containing medetomidine active substance (CP-Pharma, Burgdorf, Germany) at a dose of 40 μg/kg of live mass for sedation and analgesia of the animals.

Apaurin containing diazepam active substance (KRKA, Novo Mesto, Slovenia) at a dose of 0.3 mg/kg of live mass, combined with Propofol MCT/LCT Fresenius containing propofolum active substance (Fresenius Kabi, Linz, Austria) at a dose of 3–5 mg/kg of live mass for achieving and maintaining, as necessary, general anaesthesia.

#### 2.1.3. Antibiotic Therapy

Synulox containing amoxicillin active substance and clavulanic acid, RTU injection preparation (Zoetis, Prague, Czech Republic) at a dose of 20 mg/kg of live mass s. c. on days 1 to 14 of the therapy; after day 14, at a dose of 15 mg/kg of live mass p.o., until the skin defect was closed.

Enroxil containing enrofloxacin active substance (Krka, Slovenia) at a dose of 7 mg/kg of live mass for 30 days p. o. after a bacteriological culture test with an antibiogram was performed.

#### 2.1.4. Analgesia

Tramal containing tramadolium chloride active substance (Stada, Bad Vilbel, Germany) at a dose of 2 mg/kg of live mass administered i. v. or s. c.

Meloxidolor containing meloxicam active substance (Sevaron, Brno, The Netherlands) at a dose of 0.2 mg/kg of live mass as a primary dose, and then at a half dose for the following 14 days. First days after the therapy initiation, the administration was s. c. and then peroral.

#### 2.1.5. Wound Care

Povidone iodine (Betadine, EGIS Pharmaceuticals PLC, Budapest, Hungary) in a 4% concentration, dissolved in saline solution, used after the primary wound examination.

Betadine ointment containing povidone iodine active substance (EGIS, Pharmaceuticals PLC, Hungary), used primarily in the skin defect treatment.

Baneocin ointment containing bacitracinum zincicum active substance combined with neomycini sulfas (Sandoz, Sandoz GmbH, Kundl, Austria).

Ialugen Plus cream containing silver sulfadiazine active substance combined with hyaluronic acid (IBSA, Bratislava, Slovak Republic). Topical medication was applied twice daily for seven days, then once daily, and after 14 days every 72 h for the following 30 days. After this period, it was repeatedly applied twice daily, until the therapy was terminated. During the postoperative period, the wound was monitored in regular intervals, while the macroscopic indicators of defect healing, as well as changes in the scaffold’s surface and granulation bed, were evaluated.

Gauze materials—gauze compression bands (square gauze pad, Lohmann and Rauschcer G.mbH and Co KG, Neuwied, Germany), used as the primary layer.

Synthetic cotton (Cellona, Lohmann and Rauschcer G.mbH and Co KG, Germany), used as the secondary layer.

Self-adhesive bandage (Copoly, M+H Vet s.r.o., Opava, Czech Republic), used as the tertiary layer of the external dressing.

### 2.2. Methods

#### 2.2.1. Implant Structure Design

For both animals, the implant structures were designed by applying the same method. The goal was to create a flexible implant structure that would ensure creating an accurate and tight envelope around the defect in all planes. Flexibility was achieved through a low thickness of the implant structure, which was designed as a block size, as follows: l × w × d—100 × 50 × 0.4 mm (Figure 1).

#### 2.2.2. D Printing of the Implant Structure

The implants were printed out using the DeltiQ 2 Plus 3D printer (TriLAB, Brno, Czech Republic). Considering the thickness of 0.4 mm of the implant structure, the implant consisted of two layers only, in order to achieve its flexibility. Print parameters for the polycaprolactone polymer material are listed in the table below (Table 1).

## 3. Extensive Defect Therapy Management

This case study relates to two different patients (domestic dogs—*canis lupus familiaris*) that were referred to the Clinic for Small Animals at the University Hospital due to a failure of the existing therapy. Both patients exhibited signs of an extensive traumatic injury in the distal part of the pelvic limb, spreading from the metatarsal region to the heel joint and affecting the subdermal structures of the lateral surface of the limb. Both animals were clinically examined, while their condition was evaluated and the extent of damage to the skin and the surrounding structures was identified. The patients were monitored, the surface of the wound was mechanically cleared off contaminants, and samples were collected for microbial culture tests with an antibiogram in a certified laboratory. Following the primary examination, the wound was rinsed with a bacteriostatic and bactericidal lavage solution, and the edges of the skin defects were refreshed by excochleation. The initial antibiotic therapy included a combination of amoxicillin and clavulanic acid at an elevated dose, administered subcutaneously. The external treatment of the wound was carried out using a iodine ointment combined with an ointment base containing bacitracinum zincicum and neomycini sulfas. The wound was covered with a three-layer external dressing, i.e., a gauze material, synthetic cotton, and self-adhesive bandage. Bandages were changed in 24-h intervals. The condition of the skin defect deteriorated—persisting secernation of the exposed tissue with subsequent tissue maceration and defect extension were observed. For the purpose of the specification and identification of microorganisms, standard procedures were supplemented with an analysis of the protein spectrum of the target microorganism using the MALDI-TOF mass spectrometry. The efficacy of the antibiotic preparations against pathogens identified in a culture test was identified using fully automatic VITEK2 analysers intended for the identification of a minimum inhibition concentration of antibiotics. The results of sensitivity tests are interpreted and updated on the yearly basis, as prescribed by the European Committee on Antimicrobial Susceptibility Testing (EUCAST). Bacteriological culture tests confirmed the presence of MRSA in one patient and a highly resistant strain *Staphylococcus haemolyticus* in the second patient.

Due to such a deteriorated condition and transition to a chronic stage (Figure 2), the therapy was supplemented with a new antibiotic containing enrofloxacinum, administered once daily s. c. Ointments were replaced with a cream containing silver sulfadiazine combined with hyaluronic acid, and the wound was covered with a scaffold of the temporary implant.

The implantation of the scaffold was indicated after surface contaminants, eschars, and desiccated parts of the detached skin were cleaned off the wound. During these procedures, the animals were monitored and then put under general anaesthesia. In both patients, the mesh implant was attached directly to the surface of the damaged structures, without fixing it with a suture. The mesh was positioned so that the limitation to the patient’s mobility was minimal and exudate drainage was ensured.

After the procedure, the treated wound was covered with a surface dressing, in order to prevent the contamination of the granulation bed, formation of stagnant oedema, and iatrogenic damage. Subsequent postoperative treatment included the removal of the external dressing and control of oozing blood or wound exudate. Due to the extent of the damage, the small amount of blood that was present on the bandage several hours after the surgery was acceptable. Over the first few days, the bandage was changed twice a day; after 7 days, it was changed once a day; and after 14 days, it was changed every 72 h for the period of the following 30 days. After 44 days, the external dressing was removed. The skin defect with the temporary implant was left uncovered.

Until the defect was fully healed, a cream containing silver sulfadiazine combined with hyaluronic acid was applied to the implant surface and the surrounding tissues, instead of the bandage. The combination of these ingredients prevented the development of an infection caused by aerobic and anaerobic bacteria and supported the migration and division of epithelial cells, as well as the formation of new vessels at the wound site. Over the entire regeneration period, from the application of the temporary implant until the full healing of the defect, the sites where the implant contacted the surrounding soft structures were evaluated based on healing progress indicators, i.e., secernation, hyperaemia, oedema development, and necrotising of the defect surface. None of these indicators had been observed during the entire regeneration period, since the temporary implant was applied.

These clinical cases differed in the defect extent and the selected application technique; in one of the cases, the implant was applied and changed when the bandages were changed, and in the other case, it was left on the wound for the entire therapy duration. As early as in the first week after the application of the temporary implant, the formation of a new granulation bed and epithelisation of the wound edges were observed on the defect borders. A granulation site gradually formed where the implant was in contact with soft tissues on the defect surface (Figure 3).

Over the following weeks, defect edges exhibited gradual contraction; this was accompanied by gradual detachment of the temporary implant. The exposed edges of the implant were gradually removed as the regeneration continued until the defect was fully healed and the last segment of the temporary implant was detached.

After the full closure and regeneration of this extensive skin defect, only a minimum formation of keloid tissue was observed, compared to a common conservative management of extensive skin defects.

In both patients, the average healing time after the scaffold implantation was 68 days of the alternative therapy initiation (Figure 4).

## 4. Macroscopic Results

The applied temporary implant had several important functions in the defect regeneration process. The use of the temporary implant facilitated the bridging of the distant edges of the defect that supported the wound closure. Another equally important factor was the application of a mesh surface implant through dividing this extensive defect in smaller segments that formed the basis for newly forming cells and the growth of regeneration tissues. Bridging the defect edges and defect segmentation were the key factors that contributed to the defect regeneration and reduction of the total healing time. Thus, the administration of antibiotics become more effective, as the number of granulation deposits was several times higher, compared to the first six weeks of therapy. This could also be one of the possible reasons why this therapy was more successful than classical management. The other contributing factors included the drainage of wound exudates that are often present in chronically infected wounds. The progress of the healing process was assessed only by the evaluation of macroscopic indicators.

Since the study was not conducted on laboratory experimental animals, but on dogs with owners, histological examination could not be performed.

Monitoring was carried out during and after the regeneration process for the following processes: macroscopic indicators of the healing progress, in terms of hyperaemia; changes in the colour of adjacent soft tissues at the defect site and around it; secernation; formation of oedemas; presence of a granulation bed; epithelisation of wound edges; and wound contraction and defect closure.

After a certain period of time, since the complete closure of the skin defect, restored fur growth was observed on the newly formed skin tissue, on approximately ¾ of its total area, except for the central border where the wound edges merged.

## 5. Discussion

The active and prompt identification of a chronic lesion is one of the factors affecting the success rate of an applied therapy, together with early elimination of risk factors. The key indicator of wound transition to a chronic stage is the improper course of the wound healing process. The wound regeneration process stagnates, and it results in changes in the anatomic and functional tissue integrity. In the present case study, conservative management, with the use of surface dressing of the skin defect combined with the application of antibiotic therapy, had not brought the desired effect, even after 6 weeks [2,23].

The therapeutic application of autologous transplants to extensive skin defects does not necessarily guarantee the desired result of tissue regeneration. Therapy that comprises the use of the patient’s own autologous transplants may be accompanied by two types of complications. The first type represents the transplant-related complications caused by insufficient transplant integration in the tissues during healing. The complications directly related to the transplant also include the transplant displacement that occurs when a transplant is inadequately fixed or applied onto a movable base.

The second group of complications includes those that are directly related to the collection of an autologous transplant; they are mainly infections that result in secernation and the consequent delayed epithelisation. In some cases, it is necessary to apply a new transplant onto the site from which the autologous transplant was collected (in the case of an extensive autologous transplant) to facilitate the healing of the collection site; however, such cases are exceptional. Complications of this type also include the formation of haematoma and hypertrophic or keloid scars at the collection site [1].

The above stated facts clearly indicate that, in addition to complications associated with a transplant, there may also be complications related to the collection site—formation of secondary defects. With extensive defects, the collection site may be affected by the same complications as those that occur with the primary defect. One of the criteria that should be considered when collecting an autologous transplant is the patient’s overall health condition.

The use of additive manufacturing technology, combined with biocompatible and biodegradable materials, appears to be an optimal method for avoiding such post-transplantation complications [24], as well as the complications related to the patient’s overall condition. One of the key advantages of applying additive manufacturing technology is that it is no longer necessary to collect a transplant from the body. This prevents the formation of new, secondary defects. Another important benefit is the possibility of manufacturing temporary implants of various sizes that may be used not only for small, but also extensive defects. Sterilisation of temporary biocompatible implants reduces the risk of development and transmission of infection at the implantation site.

Selecting appropriate materials [25] is another factor that may facilitate and support the regeneration of the surrounding tissues associated with epithelisation—this will ultimately be reflected in the total time necessary for wound regeneration. Another way to accelerate regeneration and support epithelisation is the application of stem cells onto the temporary implant.

Similar cases studies may be found the paper titled “A Gelatin-sulfonated Silk Composite Scaffold based on 3D Printing Technology Enhances Skin Regeneration by Stimulating Epidermal Growth and Dermal Neovascularization”, published by a group of authors led by Si Xiong [26]. Despite the fact that the authors used different materials and applied different methods, the results of skin tissue regeneration were similar in these two cases.

Another published study, similar to the presented case study, focused on the use of a PCL scaffold combined with Juglone for tissue engineering, focusing on the skin. The study published by the team around Musa Ayran focused on the production of wound dressing with a combination of PCL and Juglone material, which is used as a bi-active material. Biocompatibility tests in this case recorded biocompatible behavior from the first day of incubation [27]. Despite the absence of the use of Juglone, the presented case study achieved comparable results during in vivo clinical application.

## 6. Conclusions

The presented study describes the use of a PCL implant in the healing process of extensive skin defects in two patients. The same two patients were initially treated with conservative management for six weeks, which did not bring the desired results. This period of therapy serves as “a control group” for the purposes of this study. With regard to the therapy of extensive skin defects, described in several case studies, the verification of this innovative therapeutic procedure on a bigger group of patients might bring clearer conclusions and results. Based on the positive results of the presented study, this therapeutic procedure might be accepted not only in the veterinary practice, but also in human medicine, in the therapy of extensive skin defects that occur, for example, with burns of various degrees, decubitus, and other injuries. The implantation of a scaffold, used as the external dressing in this case study, represents a good alternative application for a chronic extensive defect, where it was impossible to apply a skin flap or close the defect directly. Such an implantation structure, together with tissue regeneration, facilitated achieving a similar effect and accelerated the healing of the extensive skin defect. Advantages provided by this procedure include defect breathing and bridging and the formation of new foci organized on the skin; all of these factors reduce the risk of serious complications.

## Figures and Tables

**Figure 1 polymers-14-05293-f001:**
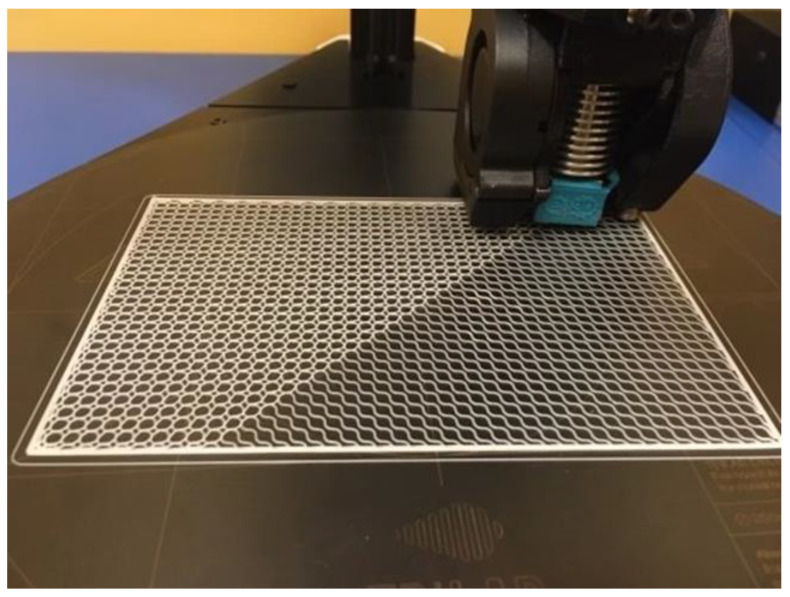
D printing process; size: l × w × d—100 × 50 × 0.4 mm.

**Figure 2 polymers-14-05293-f002:**
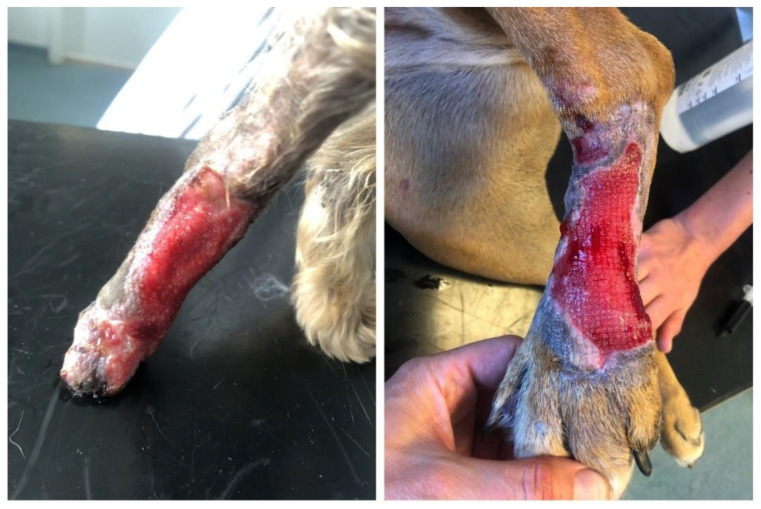
The defect prior to therapy initiation: Patient 1 on the (**left**); Patient 2 on the (**right**).

**Figure 3 polymers-14-05293-f003:**
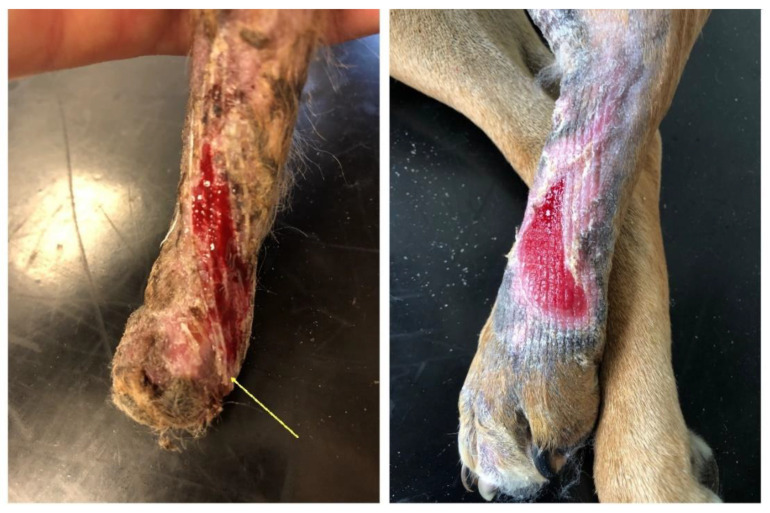
Condition after 30 days of therapy: Patient 1 on the (**left**) (the yellow arrow indicates the residual surface implant made of PCL); Patient 2 on the (**right**).

**Figure 4 polymers-14-05293-f004:**
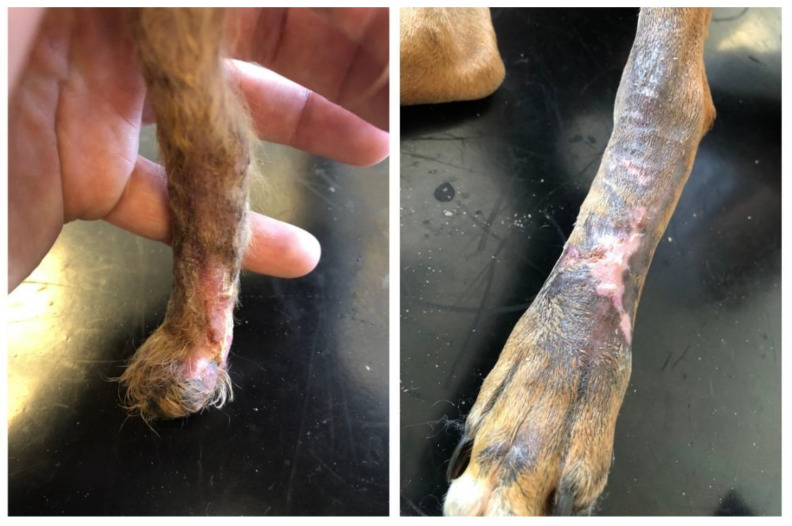
Therapy termination and closure of the extensive skin defect: Patient 1 on the left after 91 days of the surface scaffold implantation (**left**); Patient 2 on the right after 68 days of the initiation of the alternative therapy using a surface implant (**right**).

**Table 1 polymers-14-05293-t001:** Printing parameters.

Material	PCL
Nozzle temperature	100 °C
Platform temperature	45 °C
Print rate	~20 mm/s
Filament thickness	1.75 ± 0.05 mm

## Data Availability

The data presented in this study are available on request from the corresponding author.

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
