# Peer review of "Therapy of Extensive Chronic Skin Defects after a Traumatic Injury Due to Microbial Contamination Using a Surface Implant Made of a Biocompatible Polycaprolactone—A Pilot Case Study"

_polymers, 2022, doi:10.3390/polym14235293_

Round 1

Reviewer 1 Report

This is very interesting case study. It is very well written and results are easy to follow. I am happy to recommend acceptance of this manuscript even in present form. However, I'd like to make few comments:

1. Materials/methods section was a bit hard to follow, it could be improved.

2. I completely understand why you have chosen PCL to work with, but it might be good to explain that to readers in Introduction or Materials section. 

3. Do authors have further plans on trying other biomaterials in the similar study? If yes it would be good to mention that in the Conclusion.

Author Response

Dear Reviewer,

thank you for the revision of our manuscript. We hope, that you will be satisfied with our updated version.

1.-2. We updated the Introduction/Material/Methods part. We also described reasons why we chose PCL material in Discussion and Introduction.

3.Yes, we plan to conduct another similar study in the future.

Kind regards,

Authors

Reviewer 2 Report

A potentially promising paper investigating 3D printed implants for treating chronically infected wound healing in dogs. However, from reviewing the manuscript as it currently stands I can not recommend its publication.

The claims made in the abstract are too strong for the data presented. My main concerns are the use of only two animals, and the lack of controls (I understand this is difficult with using animal patients). Many additional therapies are used on the infected wounds in addition to the implant (antibiotic therapy for example). I am not currently convinced the 3D printed implant contributed as significantly as claimed by the authors. 

If the authors could enrol more patients on the study to demonstrably demonstrate the impact of the implant then this would greatly strengthen the paper for publication. For this future submission it would be interesting for the authors to characterise the implant, it is claimed it is flexible, but there is no data provided to prove this.

Author Response

Dear Reviewer,

thank you for the revision of our manuscript. We hope, that you will be satisfied with our updated version.

We have made the requested changes in the abstract. The patients included in the study were treated with antibiotics and other conservative methods for 6 weeks without success before the implantation. Only the combination of previous therapy with the implantation of a PCL implant brought the desired results. The defect was divided into smaller areas by the application of the implant, which served to create more deposits for the formation of new granulation tissue and thus accelerated healing. This could be also one of the possible reasons why this therapy was more successful than classical management.

Flexibility was achieved through a low thickness of the implant structure that was designed as a block-sized as follows: l x w x d – 100 x 50 x 0.4 mm and „Biodegradable and biobased, our PURASORB® polymers promote natural wound management and healing via safe and effective surgical and other fiber-based products. They are extremely flexible and can be used in synthetic resorbable mono- and multi-filaments. What's more, our leading polymers can be processed using a range of techniques, including extrusion, fiber-spinning, braiding, and coating processes.“ - https://www.corbion.com/en/Products/Biomedical-products/polymers-for-medical-devices (declared by the seller)

In the future, we plan to conduct a more extensive study with a higher number of patients. This study was intended to serve as a pilot study of the use of PCL implants in veterinary medicine for wound healing.

Kind regards,

Authors

Reviewer 3 Report

The attached article is easy to understand and knowledgeable for scientists to put as a reference. The conclusion also pertains to the result presented. Check minor grammar correction.

Author Response

Dear Reviewer,

thank you for the revision of our manuscript. We hope, that you will be satisfied with our updated version.

Kind regards,

Authors

Reviewer 4 Report

The article used a 3D-printed-PCL mesh as superficial wound dressing in therapy of chronic wounds, further detailed defect therapy management was described, which is quite meaningful to therapists. However, there are no any innovation of materials and no enough detailed information of PCL-implant, which are very important for the readership of the journal of Polymers. Fatally, no result of control group was clearly shown in the article. Hence, this article only can be considered in the journal of Polymers after a major revision. The detailed comments are followed:

(1)    In the introduction part, the advanced materials for the scaffold of wound dressing should be summarized here. The authors need to know why the 3D-printed PCL scaffold was chosen for the research; From the introduction, the readers would like to know what’s new in the research.

(2)    In the experimental part, the detailed parameters of PCL material and 3D-printing should be addressed, for example, the molecular weight of PCL, the pore size of 3D-printed PCL mesh. The detailed information should ensure that the experiment can be repeat by other researchers.

(3)    The control group without the PCL surface implant should be clearly shown, which will indicate the role of PCL surface implant in the therapy.

(4)    At page 6 line 220, the “left” should be “right”.

(5)    At page 7 line 229, I think “Figure 3” should be “Figure 4”.

Author Response

Dear Reviewer,

thank you for the revision of our manuscript. We hope, that you will be satisfied with our updated version.

This study was intended to serve as a pilot study of the use of PCL implants in veterinary medicine for wound healing. In the study, we described so far two cases where a PCL implant was used. Since this is a clinical study using animals owned by the owners (not experimental animals), there is no control group. However, in these patients, before the use of implants, a conservative form of therapy was applied with no result - no progressive wound healing. Only after the implantation of the PCL implant we noticed the desired results. So, in this sense, we can consider the first 6 weeks of therapy in these patients as a "control group".

Of course, after the successful proof of the positive effect of the PCL implant in the healing of extensive skin defects, we plan to conduct a more extensive study using this polymer material, possibly including a control group of animals in the study.

Correction of point 4. and 5. – done as well

Kind regards,

Authors

Reviewer 5 Report

Title: Need to include the type of material in the title

Abstract: Too general. Need to add on the problem statement and main findings.

Introduction: Problem statement is not clear. Need to include the reason PCL as the material and the differences of this material with other wound dressing materials

Materials: Section 2.1 need to be revise into proper sentences

Methods: Figure 1 : Need to include the image of the printed scaffold with the scaffold dimension

Result and Discussion:

-is there any in-vitro test had been conducted to support in-vivo study?

-need to elaborate more on results and discussion.

Conclusion: Lack of information on main findings.

Author Response

Dear Reviewer,

thank you for the revision of our manuscript. We hope, that you will be satisfied with our updated version.

Title: The type of material used is added in the title.

Introduction, material, and methods. The required information was supplemented, and the introduction expanded –the reasons why the PCL material was chosen for the study included

Material and methods part – corrected as well

Results and Discussion: information for PCL material and in-vivo studies added.

Conclusions: The conclusion supplemented by clear evidence of the positive effect of PCL implantation on the healing of extensive defects.

Kind regards,

Authors

Round 2

Reviewer 2 Report

I am happy with the changes the authors have made and would now recommend publication

Author Response

Dear Reviewer,

thank you for the revision of our manuscript. 

Kind regards,

Authors

Reviewer 4 Report

Accept

Author Response

(The authors gave the same response as above.)
